# Microwave Sintering Rapid Synthesis of Nano/Micron β-SiC from Waste Lithium Battery Graphite and Photovoltaic Silicon to Achieve Carbon Reduction

Min Zhao [1,2,3,*], Qin Chen [1,2,3], Michael Johnson [4], Abhishek Kumar Awasthi [5,*], Qing Huang [1,2,3], Weihua Gu [1,2,3], Chenglong Zhang [1,2,3], Jianfeng Bai [1,2,3], Zhen Tian [1,2,3], Ruyan Li [1,2,3] and Jingwei Wang [1,2,3,*]

[1] School of Resources & Environmental Engineering, Shanghai Polytechnic University, Jinhai Road No. 2360, Pudong New District, Shanghai 201209, China; chenqin@sspu.edu.cn (Q.C.); huangqing@sspu.edu.cn (Q.H.); whgu@sspu.edu.cn (W.G.); clzhang@sspu.edu.cn (C.Z.); jfbai@sspu.edu.cn (J.B.); tianzhen@sspu.edu.cn (Z.T.); ryli@sspu.edu.cn (R.L.)

[2] Research Center of Resource Recycling Science and Engineering, Shanghai Polytechnic University, Jinhai Road No. 2360, Pudong New District, Shanghai 201209, China

[3] Shanghai Collaborative Innovation Centre for WEEE Recycling, Shanghai Polytechnic University, Jinhai Road No. 2360, Pudong New District, Shanghai 201209, China

[4] Department of Electronic & Computer Engineering, University of Limerick, V94 T9PX Limerick, Ireland; Michael.Johnson@ul.ie

[5] School of the Environment, Nanjing University, 163 Xianlin Road, Qixia District, Nanjing 210023, China

* Correspondence: zhaomin@sspu.edu.cn (M.Z.); abhi28@nju.edu.cn or abhishekawasthi55@ymail.com (A.K.A.); jwwang@sspu.edu.cn (J.W.)

**Abstract:** The paper describes one promising method and approach for the recycling, reuse, and co-resource treatment of waste photovoltaic silicon and lithium battery anode graphite. Specifically, this work considers the preparation of nano/micron silicon carbide (SiC) from waste resources. Using activated carbon as a microwave susceptor over a very short timeframe, this research paper shows that nano/micron β-SiC can be successfully synthesized using microwave sintering technology. The used sintering temperature is significantly faster and more energy-efficient than traditional processes. The research results show that the β-SiC particle growth morphology greatly affected by the microwave sintering time. In a short microwave sintering time, the morphology of the β-SiC product is in the form of nano/micron clusters. The clusters tended to be regenerated into β-SiC nanorods after appropriately extending the microwave sintering time. In the context of heat conversion and resource saving, the comprehensive $CO_2$ emission reduction is significantly higher than that of the traditional SiC production method.

**Keywords:** waste lithium-ion battery; photovoltaic silicon; graphite; SiC; microwave; carbon emission reduction

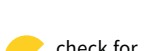



## 1. Introduction

To address global warming, pollution, and other environmental issues, mankind is striving to reach zero carbon emission levels by the middle and end of the 21st century. To achieve this, the development of renewable energy solutions, such as solar energy, will be pivotal to this process. Solar energy system installations are already increasing; it is predicted that the worldwide solar installed capacity will range from 2.48 TW in 2020 to 8.5 TW by the year 2050, which will offer significant (between 2.5–25%) of the global electricity demand [1,2].

Presently, at a global scale, for example, in terms of the application of photovoltaic (PV) power generation, China's installed total PV capacity has reached 186 GW in the first half of 2019, accounting for most of the global installed PV capacity [3]. Studies predict that this will lead to an unprecedented increase in waste PV modules in China by 2025, expected to culminate in 56.5~62.6 GW of waste PV modules by 2050, aggregating to a total



of 702.7 GW [4]. Given that the average PV panel's lifetime is 25 years, this means that the worldwide solar PV waste generated will be between 4% and 14% of the total e-waste generated globally by 2030, increasing to over 80% (around 78 million tons) by 2050 [5].

PV modules are made up of aluminum and glass, including several harmful trace elements, such as antimony, lead, and cadmium [6–8]. Some component materials, such as silicon, silver, and copper, are also present and are currently widely recycled and recovered [9–12]. Recycling is therefore the preferred and most effective means of processing these waste PV modules and recovering the constituent materials [12]. To effectively recycle the panels, however, as the core material of solar cells, the reuse of high-purity silicon in these solar photovoltaic cells and panels is crucial.

The use of electric batteries and the corresponding reduction in traditional fuel consumption are other strategies employed to reduce carbon emissions and address global environmental pollution issues. Lithium-ion batteries (LiBs) have become widely used in mobile phones, electric vehicles, and several different electronic consumer items. The World Bank report suggests that the worldwide ratio of mobile phone subscriptions touched 1.04 per person in 2017, with an unending rising trend [13]. In this context, China is the world's major producer and consumer, and progressive data display the figure of obsolete mobile phones generated in 2020 to reach 877.8 million in China alone [14], while estimates indicate that the cumulative total of electric vehicle sales volume is expected to reach 465 million in the United States by 2050 [15]. However, LIBs have a limited lifetime, sometimes as short as 3–5 years [16]. In 2020, the estimated prediction indicates that there will be a significant volume (13,828 tons) of waste LIBs in the EU that will be processed [17], with China expected to recycle >5 billion tons of LIBs [18].

Waste LIBs hold great values of heavy metal elements, such as cobalt, nickel, manganese, and copper; thus, there is a great loss of secondary resources, and this constitutes a serious environmental risk if haphazardly disposed of. One study reported that 4000 ton of waste LIBs contained 1100 tons of heavy metals, as well as 200 tons of toxic electrolytes [19]. Such materials, present in such quantities, threaten both the environment and humans if not properly processed and disposed of [20].

In recent years, the recycling of spent LIBs has attracted significant attention. However, recycling of the electrode materials has predominantly focused on the cathode materials, owing to the highly valuable metals and elements present therein [21–24]. Less attention has been paid to recycling the anode material, as it is composed mainly of graphite, which has a lower added value and is more difficult to regenerate. A solution that would allow for the efficient and cost-effective recycling of graphite is necessary to further the recycling and processing of spent LIBs. One possible solution to these recycling concerns is silicon carbide (SiC). SiC ceramic is potentially one of the highest temperature structural materials in existence, owing to its excellent properties such as high strength, high thermal conductivity, hardness, and excellent corrosion resistance, electromagnetic wave absorption materials [25,26].

The work defined in this article addresses the shortcomings of recycling solar PV and LIBs by focusing on a new waste crystal silicon and waste graphite utilization based on microwave sintering technology, using solar PV and LIBs as source materials. Waste crystal silicon wafers are separated from waste solar PV panels and purified for use in such a process. Graphite can also be separated from spent LIBs for use in this process. These two sources of bulk electronic waste (e-waste) were used to synthesize β-SiC powder by microwave rapid sintering. The proof of concept of this technical solution for recycling LIB and solar PV e-waste is also provided.

## 2. Materials and Methods

The waste crystalline silicon of solar photovoltaic cells and waste graphite materials are the research objects. Photovoltaic cell silicon wafers were burned to remove organic matter at high temperature (450 °C, 40 min), and metal was removed by chemical leaching. Firstly, treat with 30% NaOH solution for 60 min, then wash with distilled water, and treat

with 5.0 mol/L $HNO_3$ solution for 50 min. The chemical reagent immersion was completed in the ultrasonic and shaking process, and the resulting silicon was ultrasonically cleaned. The graphite comes from the negative electrode of dismantled waste lithium-ion batteries, which are purified by a chemical method, the LIBs material is subjected to alkali leaching and acid leaching processes, and the valuable metal ions exist in the filtrate. The filter residue is coarse graphite product. The coarse graphite product is screened, and air is selected to remove large particles of Cu and diaphragm materials to obtain preliminary purified graphite. Then, it is subjected to secondary acid leaching to remove small particles of metals such as Cu, etc. Finally, the purified graphite is washed to neutral with distilled water and dried at high temperature in the air. Additional components such as (chemically pure) granules of activated carbon (AC) (particle size 0.5–1 mm) were acquired from Sinopharm Chemical Reagent (SCRC) Co. Ltd. (Shanghai, China).

The synthesis of the material was achieved using a microwave reactor. The reactor is a rectangular multimode microwave cavity rotary reactor with 700 W power input and a microwave reactor frequency of 2.45 GHz. The rotary reactor used in this study was modified from a microwave oven (Whirlpool China Co., Ltd. Hefei, China). This rotary microwave reactor can improve the uniformity of microwave irradiation. An optical pyrophotometer (Shanghai Automation Instrumentation Factory, WGG2-201) was used to measure the working temperature.

The initial stage of the procedure involves crushing crystalline silicon waste. Stoichiometric amounts of lithium battery graphite and Si (1:1 molar ratio) were then added to this material using a planetary ball mill and milled for 5 h at 400 rpm, after which the milled powder was cold pressed in an 8-mm pellet die (5 tons), which were then pressed into pellets (0.3–0.35 g each). Granular AC was first placed in an open small quartz tube, and the pressed pellets were subsequently placed in the center of the same quartz tube, and granular AC was again placed in the small quartz tube, which was enclosed in an aluminum tube. Finally, the quartz tube was placed in the center of the microwave reactor, and the microwave sintering process was initiated. At the same time, the sintering temperature was measured and recorded using an optical thermometer through the visible light intensity of the quartz glass tube wall.

All the planning and experiments were conducted in open air. The specific experimental conditions for all samples are presented in Table 1.

**Table 1.** Chosen microwave reactor samples in addition with reaction conditions.

| Sample number | $A_1$ | $A_2$ | $A_3$ | $A_4$ | $A_5$ | $A_6$ | $A_7$ | $A_8$ |
|---|---|---|---|---|---|---|---|---|
| Sintering time(min) | 5 | 7 | 9 | 11 | 13 | 15 | 17 | 19 |

A Bruker D8 Advance (Cu Ka1 radiation, $\lambda$ = 1.54056 Å) was used for the X-ray diffraction (XRD) measurements. Using the EVA5.0 software, the initial data were used to recognize product phases in the context of the COD database. The Rietveld refinement method was based on the XRD data for each dataset (Topas6.0 and EVA5.0). Scanning electron microscopy (SEM) (Phenom Prox) was used to investigate the morphology and elements of the sample, and the samples were gold-plated to enhance the sample conductivity before SEM analysis. A laser scattering particle size distribution analyzer (Horiba LA-960) was used to determine the particle size of the SiC samples, with the dispersing medium being distilled water and SiC (refractive index, 2.64). The Raman spectra were obtained using a Jobin Yvon XploRA confocal microRaman system (532 nm laser, power ~80 mW, acquisition time 1 s).

Determining the metal elements content in the raw materials with ICP (Thermo iCAP 7000 SERIES), The 18 metal mixed standard samples were used, and the standard solution for the standard curve was diluted with 5% $HNO_3$, and the sample was washed 3 times and kept for 30 s.

## 3. Results

Using a microwave rotary reactor (700 w input), micron-sized β-SiC from the recycled silicon and graphite was successfully synthesized. In the initial synthesis experiments, the microwave sintering time was 5 min. At this point, the main peaks are the silicon and graphite diffraction peaks (Figure 1A$_1$), with the peaks of β-SiC being very low, indicating that there was still a large amount of elemental silicon and graphite in the sample, which had only been partially β-SiC synthesized (Figure 1A$_1$).

From the A$_2$–A$_8$ samples (Figure 1), as the microwave sintering time increased, the main diffraction peaks were β-SiC. During the microwave sintering, it is found that if the samples are not placed directly in the quartz tube maximum temperature region, it will be resulting in some or part of the sample not being successfully synthesized. Therefore, not all areas will be able to attain the SiC-synthesized temperature in a short time during the microwave sintering process. This problem was avoided in these experiments owing to the user's experimental experience. The synthesis of SiC is therefore dependent on a uniform and sufficient temperature, and once this has occurred, the rapid synthesis of SiC is guaranteed using this setup.

Referring again to Figure 1, the crystal structure of the resultant β-SiC was not influenced by the microwave sintering time, and there were no obvious variances among according to the judged from the XRD curves of A$_2$–A$_8$, all of which were also successfully recognized as β-SiC. To synthesize SiC products, the synthesis of the SiC products was completed after sintering for 7 min. A small peak (2θ = 33.6°) ahead of the maximum peak was attributed to stacking faults (marked with ◆) on the 111 plane in β-SiC [27].

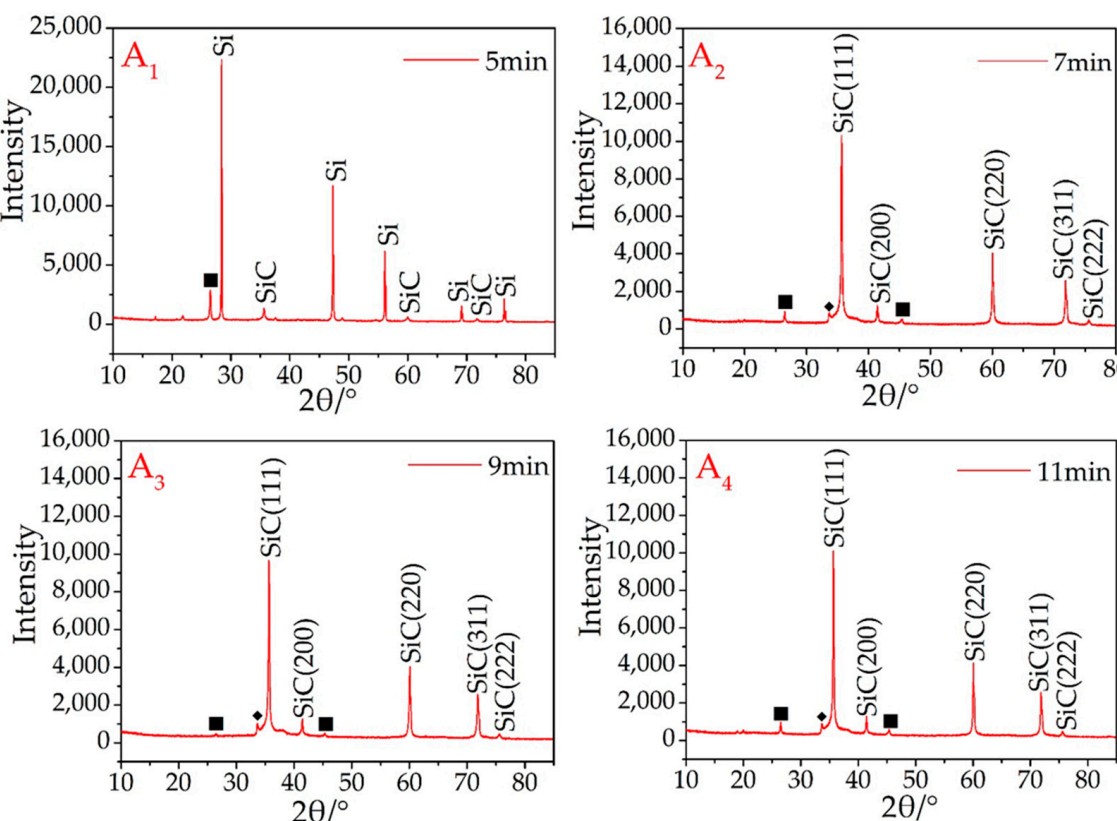

**Figure 1.** *Cont.*

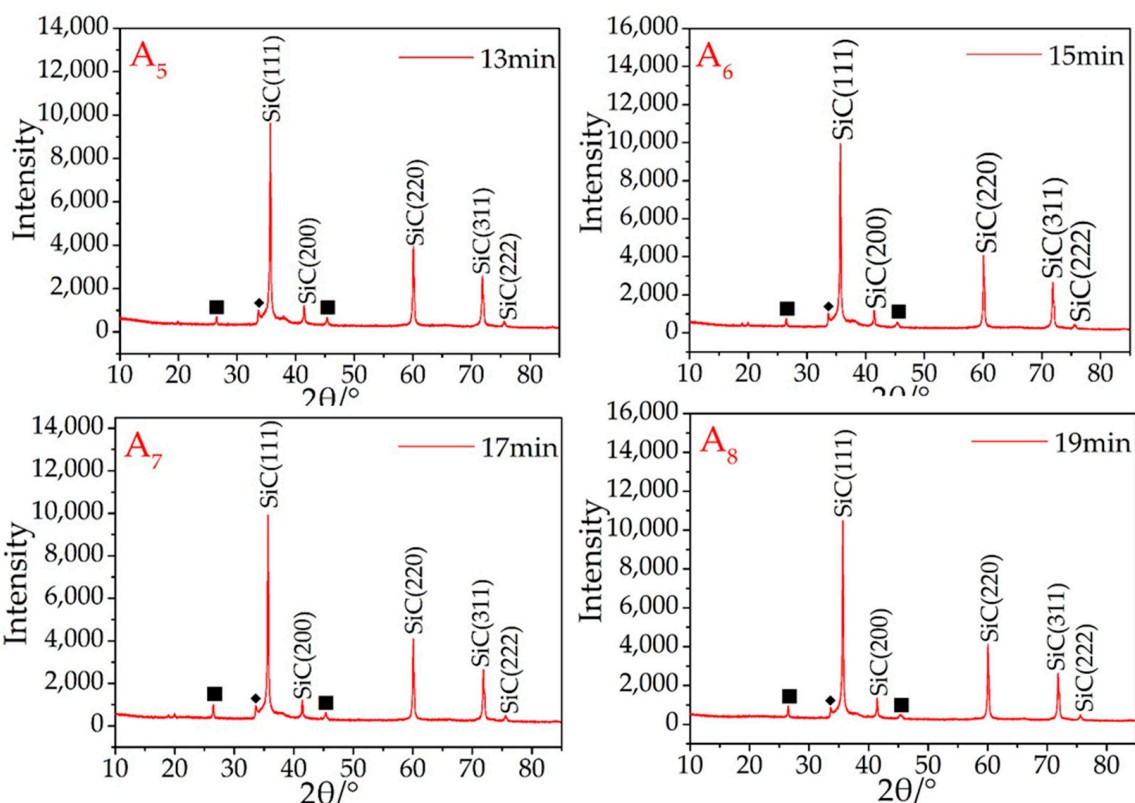

**Figure 1.** XRD patterns of synthesized β-SiC in the microwave reactor. Graphite (■).

There is always a weak graphite peak present in the results (Figure 1A$_2$–A$_8$), owing to two different factors. The first is the use of granular AC as a sintering aid. The use of Si and AC to synthesize SiC is easier than using graphite and Si. The synthesis temperature for such a process is also lower, which causes part of the silicon (usually the Si in contact with or in proximity to the AC) in the sample to prematurely combine with the AC. In addition, during the experiment, it was found that a small amount of SiC was formed on the surface of the granular AC around the sample. This reaction consumes silicon and leads to an excess of graphite.

Second, owing to the limitations of the process (such as insufficient mixing of raw materials), silicon cannot be combined at 100%. Because silicon is partially volatilized under high-temperature conditions, the volatilization ability of graphite is much lower than that of silicon at the same temperature. Therefore, after sintering is completed, some graphite will always remain.

The maximum temperature reached 1600 °C in this microwave sintering process, and the temperature gradient declined and tended to reach a lower stability after extending the heating time (Figure 2). When the process temperature reached 1450 °C (Figure 2), the key reaction was nearly complete (Figure 1A$_4$). The temperature of 1600 °C is higher than 1410 °C (silicon melting point) but far lower than 2355 °C (silicon boiling point) and 3727 °C (carbon melting point), and the SiC synthesis mechanism system is a liquid–solid reaction. However, there are a large number of "hot spots" that are difficult to observe in the microscopic state during the microwave sintering process. The actual temperature is therefore much higher than the temperature measured in this research, which is sometimes greater than the boiling point of silicon, meaning that a gas–solid reaction occurs.

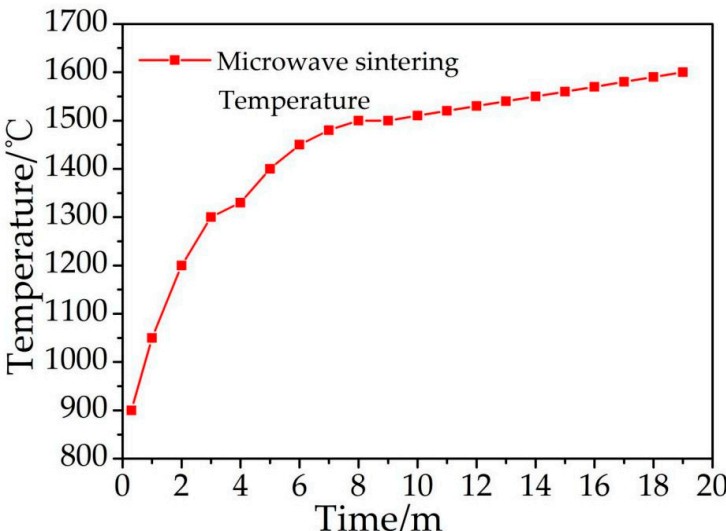

**Figure 2.** The temperature curve of microwave sintering.

The zinc blend structure as a starting model and based on the $A_8$ cell parameters, EVA5.0, was used to calculate the content of β-SiC. The results of the Rietveld refinement and quantitative analysis are shown in Table 2 and Figure 3.

**Table 2.** Crystallographic data from Rietveld refinements against XRD data.

| Sample | $A_2$ | $A_5$ | $A_8$ |
|---|---|---|---|
| Phases, wt% | β-SiC:97.8%; | β-SiC:98.3%; | β-SiC:97.1%; |
| Z | 4 | 4 | 4 |
| α-Parameter/Å | 4.329 | 4.341 | 4.329 |
| Unit cell vol/Å3 | 81.10 | 81.81 | 81.10 |
| Crystal density, $(g/cm^{-3})$ | 6.235 | 6.181 | 6.232 |
| Rp | 2.93 | 2.88 | 2.93 |
| Rwp | 3.92 | 3.87 | 3.95 |
| Rexp | 3.88 | 3.79 | 3.88 |

Space group: Cubic, F-43m (216); Si 4a (0, 0, 0), C 4c (3/4, 3/4, 3/4).

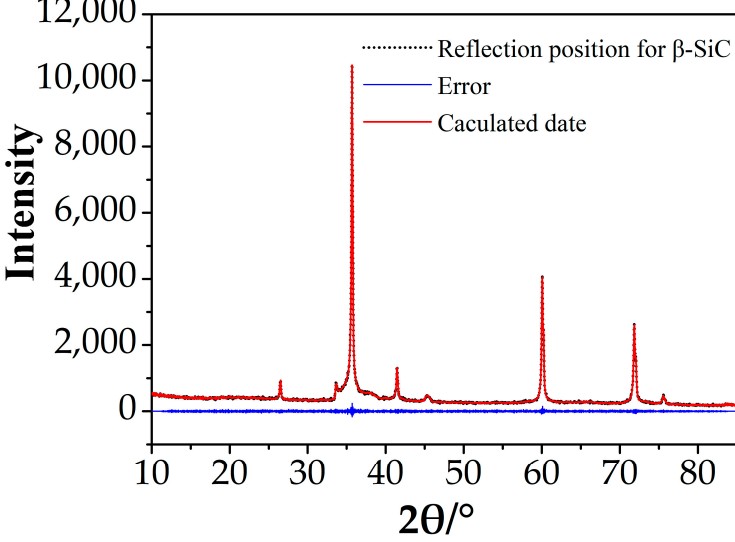

**Figure 3.** Profile plot for Rietveld refinement against XRD data of $A_8$.

The mechanism of SiC synthesis is as follows:

$$Si(l) + C \rightarrow SiC(s) \tag{1}$$

$$Si(g) + C \rightarrow SiC(s) \tag{2}$$

The stoichiometric ratio of the raw materials was Si:C = 1:1. The content of β-SiC reached 98.3% (Figure 4c; Table 2, $A_5$), in connection with reaction (1), as well as with no carbon being emitted directly, and SiC formed on the AC surface (near the pellets), in agreement with reaction (2).

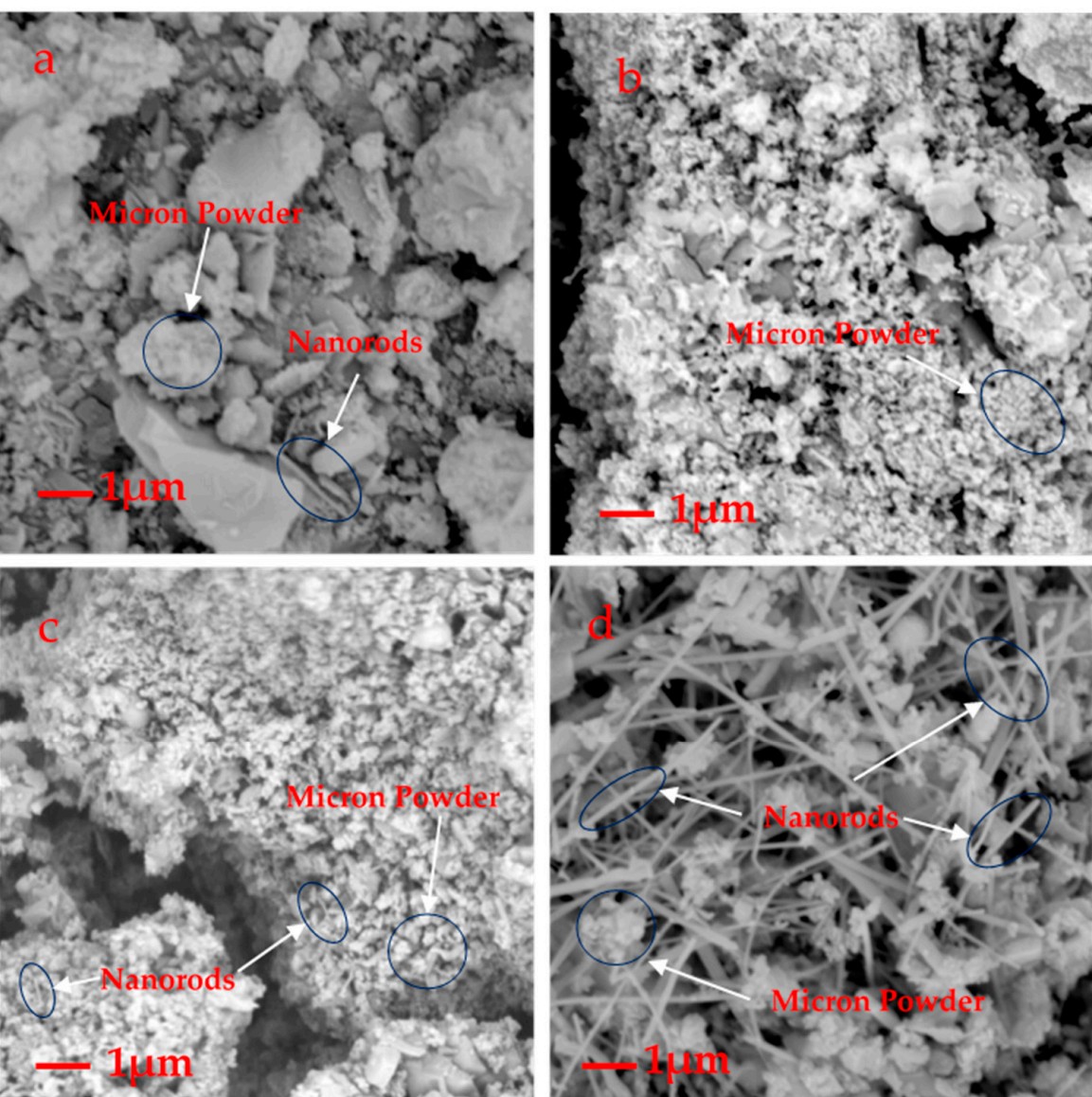

**Figure 4.** SEM micrographs of $A_2$ (**a**), $A_3$ (**b**), $A_5$ (**c**), $A_8$ (**d**).

The laser diffraction analysis showed that the sizes of the β-SiC cluster particles were proportional to the microwave sintering time. This research found that the microwave sintering time is typically 15–19 min, after which the β-SiC particles begin to grow. After approximately 4 min, D50 (the median particle size) grows from 11.963 to 59.408 μm (Figure 5, Table 3). Combined with the analysis of the SEM image (Figure 4), a large number of β-SiC particles changed from the cluster state to the nanorod state at this stage.

The amount of β-SiC nanorods formed before the microwave sintering time of 15 min was small.

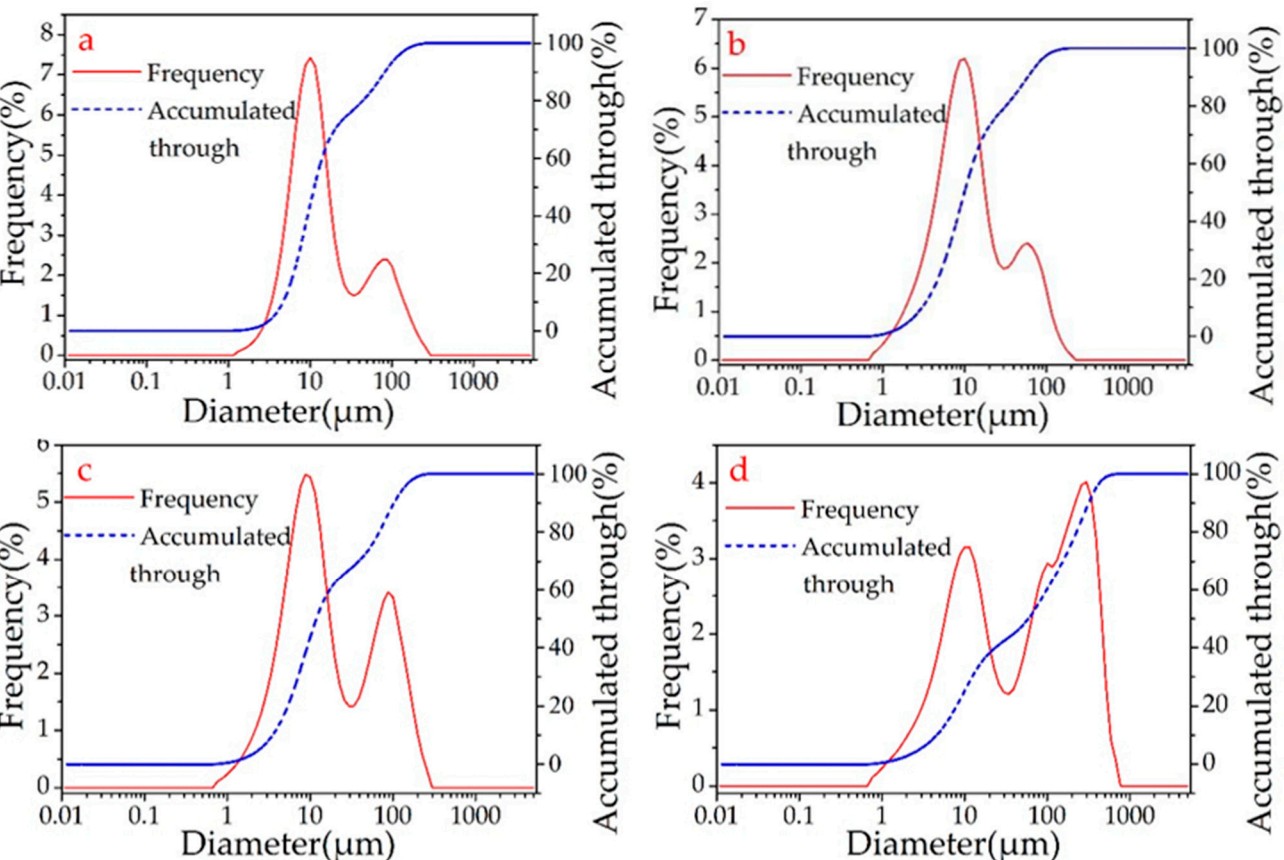

**Figure 5.** Diagram of SiC particle size distribution detected by laser particle analyzer of A$_2$ (**a**), A$_5$ (**b**), A$_6$ (**c**), A$_8$ (**d**).

**Table 3.** Statistics of particle size distribution of β-SiC determined by laser diffraction method.

| Sample | A$_2$ | A$_5$ | A$_6$ | A$_8$ |
|---|---|---|---|---|
| D50 (μm) | 11.296 | 9.972 | 11.963 | 59.408 |
| Mean size(μm) | 28.138 | 20.941 | 35.384 | 114.744 |
| Geometric mean size (μm) | 14.789 | 11.373 | 16.065 | 42.578 |
| D10(μm) | 4.680 | 3.032 | 3.528 | 4.608 |
| D90(μm) | 82.590 | 58.987 | 103.243 | 318.899 |

β-SiC mainly exists in the cluster state, which extends the microwave sintering time. After this, the β-SiC nanorods begin to form in several minutes, which can preliminarily indicate that β-SiC nanorods are based on the successful synthesis of β-SiC cluster particles. Then, the β-SiC clusters are further decomposed, and through gas–solid and gas–gas reactions, β-SiC nanorods are formed. The main cause of this is the big number of "hot spots" appearing in the microwave sintering system. The gas phase material in the sublimation growth process is usually rich in Si, and the unevaporated source becomes richer and richer in C, leading to graphitization of the raw material. The synthesis mechanism of the SiC nanorods is shown in Figure 6.

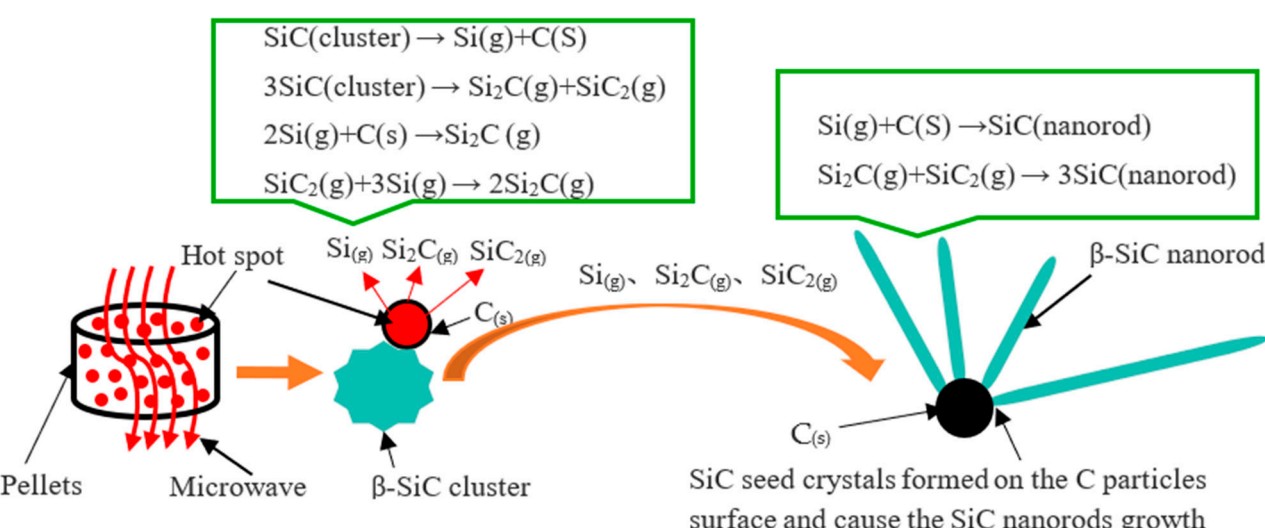

**Figure 6.** Schematic of the generation of β-SiC nanorods based on microwave hot spots.

It can be seen from the SEM that the majority clusters were bigger than 0.1 μm, consistent with their micron size; the micron particles displayed a flaky distribution or irregular mass (Figure 4a–d). This could be due to the inconsistent distribution size of the early broken waste silicon particles. As shown in Figure 4b,c, some SiC clusters were nano grade, showing the shape of the nanoclusters. Very small amounts of nanorods were found, but the majority of β-SiC clusters belonged to the micron size. In Figure 4d, with the increase in sintering time, there is a tendency for a large number of nanorods to be generated in some regions of the sample.

The elemental Si and C peaks demonstrated in the EDS result (Figure 7), and it can be inferred that the A8 sample was composed of SiC.

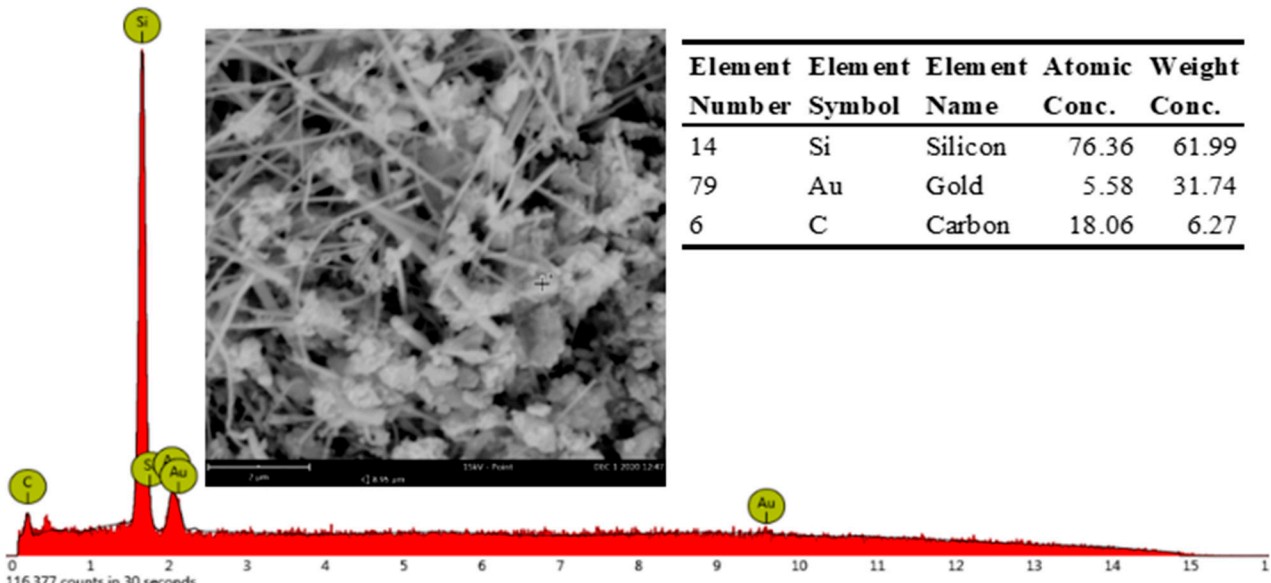

**Figure 7.** Typical EDS spectra obtained with A$_8$ sample.

It can be observed from the Raman spectrum (Figure 8) that the broadening peaks with their maximum positions at 788 (point c) and 886 cm$^{-1}$ (point d), owing to the SiC transversal optic (TO) and longitudinal optic (LO) modes [28–30]. The sample showed two additional broad peaks (Figure 8) at 521–587 cm$^{-1}$ as well as a weak shoulder (Figure 8,

point b). This may be attributed to the acoustic phonon excitation modes (TA, LA) of SiC crystallites or stacking faults [30–32].

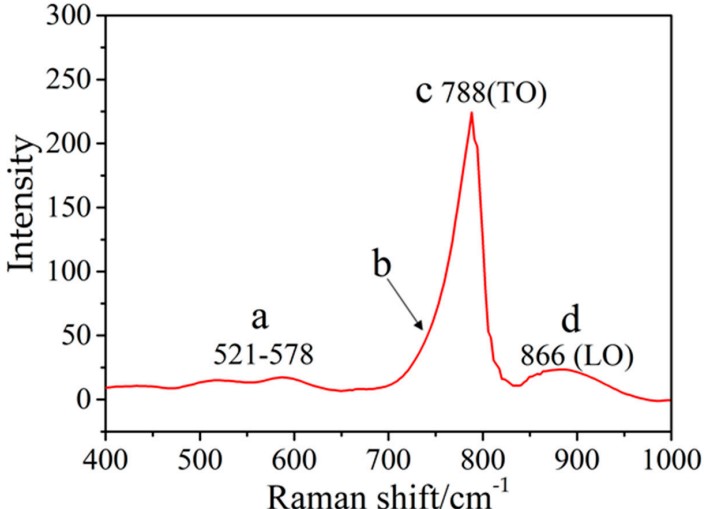

**Figure 8.** The Raman spectrum of $A_5$.

The total content of metal elements in waste photovoltaic crystalline silicon is generally low (Table 4), whereas the iron content is high. The main reason for this is the wire-cutting process for the silicon ingot. Owing to high-speed friction cutting, some iron filings are embedded in the crystalline silicon, which has a relatively limited impact on the quality of SiC. This can reduce the exploitation of natural quartz sand mining and realize energy-saving and environmental protection benefits, as discussed later.

**Table 4.** Data statistics of metal elements of waste solar PV.

| Metal Name | Fe | Al | Ni | Co | Cu | Mn | K | Li | Cd |
|---|---|---|---|---|---|---|---|---|---|
| Content (%) | 0.625 | 0.012 | 0.002 | 0.005 | 0.187 | 0.029 | 1.561 | / | / |
| Metal Name | Zn | Mg | As | Sr | Cr | Ba | Sn | Zr | Na |
| Content (%) | 0.001 | 0.001 | / | / | 0.012 | 0.001 | / | 0.479 | 1.120 |

During the recycling of waste graphite, some residual materials remained in the graphite (Table 5). The total content of metal elements in waste graphite is generally low, but the content of alkali metal elements is high. The main reason is that alkali metals are used to adjust the pH value of the recovery procedure of waste lithium-ion battery cathode and anode materials, and in the extraction process, they are used to recover cathode materials. Some of these alkali metals remain in the graphite. The presence of Zr is due to its natural occurrence with graphite minerals. Al, Cu, and Co are widely used in the manufacture of lithium-ion batteries. However, the existence of all these residual metal elements is beneficial; it is conducive to the formation of SiC nanowires/nanorods [33–38].

**Table 5.** Data statistics of negative graphite metal elements of lithium battery.

| Metal Name | Fe | Al | Ni | Co | Cu | Mn | K | Li | Cd |
|---|---|---|---|---|---|---|---|---|---|
| Content (%) | 0.100 | 0.831 | 0.266 | 0.572 | 0.255 | 0.084 | 3.815 | 0.097 | / |
| Metal Name | Zn | Mg | As | Sr | Cr | Ba | Sn | Zr | Na |
| Content (%) | 0.006 | 0.010 | 0.007 | 0.003 | 0.017 | 0.010 | 0.001 | 1.73 | 2.101 |

The spherical structure is primarily composed of C and a small amount of O. Additionally, there are many spherical graphite in the anode materials of lithium batteries

(Figure 9). The XRD patterns of graphite show that the impurities in the graphite crystal are very low, mainly graphite crystals (Figure 10) and the photovoltaic silicon is still an ideal high-purity crystal material. This shows that graphite separated from waste lithium batteries and the photovoltaic silicon separated from waste solar photovoltaic cell can be the ideal carbon raw material for the synthesis of SiC, as it can reduce the consumption of coal forest resources and quartz sand mineral.

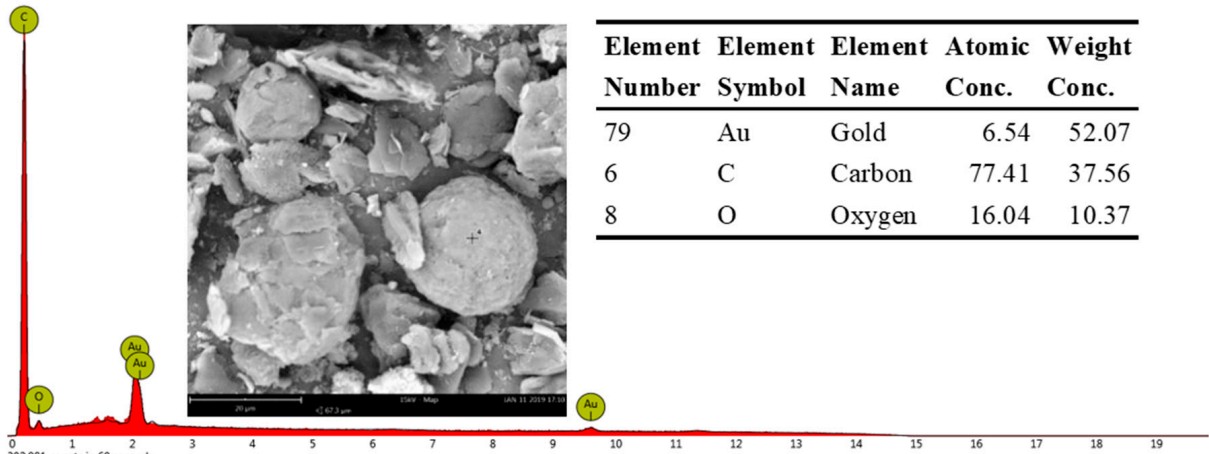

| Element Number | Element Symbol | Element Name | Atomic Conc. | Weight Conc. |
|---|---|---|---|---|
| 79 | Au | Gold | 6.54 | 52.07 |
| 6 | C | Carbon | 77.41 | 37.56 |
| 8 | O | Oxygen | 16.04 | 10.37 |

**Figure 9.** Typical EDS spectra of graphite in anode materials of lithium batteries.

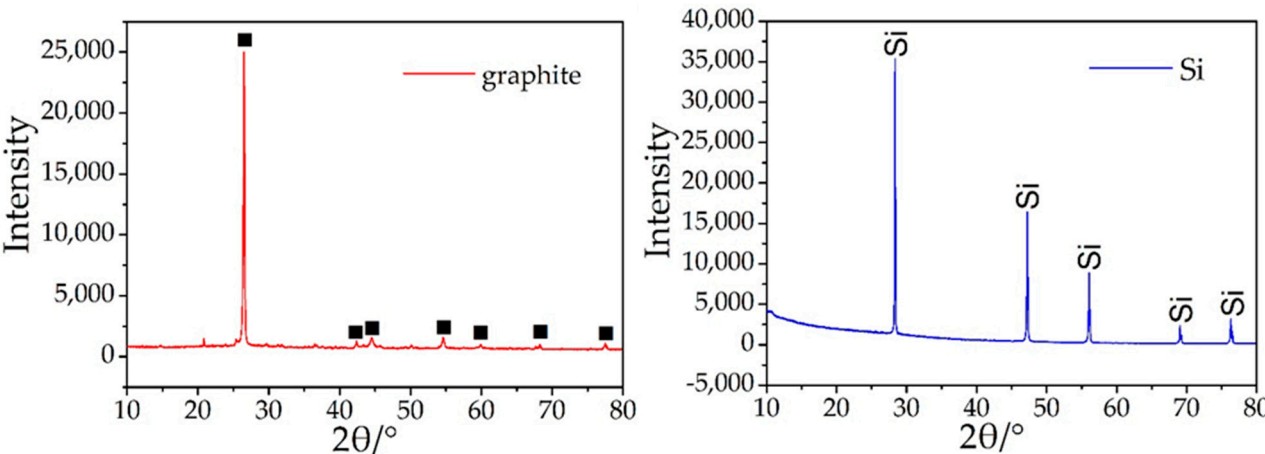

**Figure 10.** XRD patterns of graphite and photovoltaic silicon.

In this study, the SiC synthesis temperature was approximately $1550 \pm 50\,°C$ (Figure 3), much lower than those of the traditional process:

$$Q = Cm\,(T - T_0) \tag{3}$$

Using Equation (1), the heat capacity (C) of Si is 700 J/kg·K, $SiO_2$ is 800 J/kg·K, carbon is 840 J/kg·K, and graphite is (710 J/kg·K). The $T_0$ is initial temperature, which was set at 25 °C, and T is the final temperature set at 1550 °C (Si+graphite) and 2300 °C ($SiO_2$+C). Theoretical calculation results indicated that to yield a similar amount (m) of SiC, the energy consumption (Q) amount using carbon and $SiO_2$ as the raw material was approximately 3.62 times that of Si and graphite, as used in this method, with corresponding $CO_2$ releases scaled around 72% using this new method:

$$SiO_2 + 2C \rightarrow SiC + CO_2 \uparrow$$

In the field of industrial production of SiC, quartz sand ($SiO_2$) is one of the main raw materials for production, and it can be determined from the above chemical equation as follows, using a raw material such as quartz sand to produce 1.0 ton of SiC, and 1.1 tons of $CO_2$ emissions directly, while using silicon as a raw material, the direct emission of $CO_2$ is zero. The graphite and silicon can be converted for reducing the consumption of forest wood/coal and quartz sand mining, if 1.0 ton waste solar photovoltaic silicon material or 1.0 ton graphite are reused, 2.14 tons of quartz sand mining, or 1.0 ton wood/coal carbon can be saved, and the energy consumption of quartz sand/coal minerals from mining, crushing, and other processes will be saved, the forest resources can be saved.

## 4. Conclusions

This study presents a promising solution for waste silicon from solar PV panels and waste graphite (anode) in lithium-ion batteries. Both of these materials are relatively pure and have been successfully shown to function as raw materials. In particular, this paper describes the successful synthesis of high-purity β-SiC from such recycled materials.

Temperature was the primary factor influencing the fast synthesis of this micron β-SiC, with sintering time being the other important key factor for the formation of SiC nanorods. The optimal sintering synthesis time for the process was found to be ~7 min, with a large number of SiC nanorods forming past this sintering time.

This study has shown that it is possible to successfully produce micro-β-SiC in a multimode microwave reactor using such wafers. The simplified process involved is much more time-and cost-efficient than those currently employed. This, in turn, can promote the fluidization design process. In terms of heat conversion and resource saving, the $CO_2$ emission reduction was significant compared with the traditional SiC production method. This research provides an ideal resource treatment method for two bulk electronic wastes, with the resulting Nano/Submicron β-SiC expected to find use in manufacturing processes for new solar silicon wafers and other industrial products that require polishing treatment.

**Author Contributions:** Conceptualization, manuscript writing, funding acquisition, software analysis, M.Z.; project fund needs, provide materials, and part data provider, Q.C.; review and editing, M.J.; review, editing, and revising, A.K.A.; data provider, Q.H.; data provider, W.G.; funding acquisition, data provider, and guidance, C.Z.; research design and guidance, J.B.; software analysis, Z.T.; research guidance and provided materials, R.L.; funding acquisition, project administration, and provide materials, J.W. All authors have read and agreed to the published version of the manuscript.

**Funding:** This research was funded by the National Key R&D Program of China (No. 2018YFC1902303), the Innovation fund of Shanghai Polytechnic University (EGD20XQD01), and the Gaoyuan Discipline of Shanghai e Environmental Science and Engineering (Resource Recycling Science and Engineering).

**Acknowledgments:** Thanks to Ganzhou Haopeng Technology Co., Ltd., Jiangxi Province, China for providing the graphite materials of lithium-ion batteries. Abhishek Kumar Awasthi and Michael Johnson are thankful to the School of the Environment, Nanjing University, Nanjing China and the Department of Electronic and Computer Engineering, University of Limerick, Limerick, Ireland for their support, respectively.

**Conflicts of Interest:** The authors declare no conflict of interest.

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
