# Peer review of "Microwave Sintering Rapid Synthesis of Nano/Micron β-SiC from Waste Lithium Battery Graphite and Photovoltaic Silicon to Achieve Carbon Reduction"

_sustainability, doi:10.3390/su132111846_

Round 1
Reviewer 1 Report
Dear authors
I really appreciate your work. The recycling/reuse of waste photovoltaic silicon and lithium battery anode graphite to produce SiC, using the microwave technique is an interesting approach. The used characterization techniques look appropriate.
Below there are some comments that allow the authors improve the paper.
Kind regards
Abstract
|
Please review the sentence: “Specifically, this work considers the preparation of nano/micron silicon carbide (SiC) from these otherwise waste resources.” |
Introduction |
Please review the references numbers, they should appear by order of appearance in text |
Pag. 2 Methods and Materials |
“Photovoltaic cell silicon wafers are burned to remove organic matter at high temperature” – please indicate the temperature “metal is removed by chemical leaching” – please describe the procedure |
Pag 3. |
indicate AC (c.p., 0.5-1 mm) meaning |
Table 3 |
The values from sample A2 and A5 looks contradictory. Are them ok ? |
Pag 6 Pag 9 |
Formation issues: Legend of figure 3 Legend of Table 4 |
Pag 8 |
Please verify the formation, it is not possible read the legend |
Pag. 10 |
Table 4 and 5- please improve the layout, somethings are in bold and others not About legend, why the authors write: “ICP analysis data Statistics of negative graphite metal elements of lithium battery” ? I could understand if an uncertainty and confidence level were presented but was not the case. |
Pag. 11 |
Please review the sentence: “In this study, the SiC synthesis temperature was approximately 1550±50 °C (Figure 3. PV panels and waste graphite) are much lower than those of the traditional process.” |
Pag. 11 |
Can you please indicate a reference to justify the 72% CO2 reduction, please: “…as used in this method, with corresponding CO2 releases scaled around 72% using this new method.” |
Author Response
Abstract
(1) Please review the sentence: “Specifically, this work considers the preparation of nano/micron silicon carbide (SiC) from these otherwise waste resources.”
Thank you for your suggestion. We updated according to your suggestions.
(2) Introduction Please review the references numbers, they should appear by order of appearance in text.
Thank you for your suggestion. We updated according to your suggestions, and please see the revised version.
(3)pag. 2,Methods and Materials “Photovoltaic cell silicon wafers are burned to remove organic matter at high temperature” – please indicate the temperature, “metal is removed by chemical leaching” – please describe the procedure
Thank you for your suggestion. We made modifications and additions according to your suggestions, please see the red number part.
(4)Pag. 3. indicate AC (c.p., 0.5-1 mm) meaning
Thank you for your suggestion. We made modifications and revisions according to your suggestions.
(5) Table 3 The values from sample A2and A5 looks contradictory. Are them ok?
Thank you for your suggestion. We checked the original data again and confirmed that there is no problem. The conditions of microwave synthesis of different samples cannot be completely the same, which will produce such abnormal data.
(6) Pag. 6,Pag 9Formation issues: Legend of figure 3
Thank you for your suggestion. We modified it according to your suggestion.
(7) Legend of Table 4
Thank you for your suggestion. We modified it according to your suggestion.
(8) Pag. 8 Please verify the formation, it is not possible read the legend
Thank you for your suggestion. We have modified it according to your suggestion.
(9) pag.10 Table 4 and 5- please improve the layout, somethings are in bold and others not
Thank you for your suggestion. We have modified it according to your suggestion.
(10) About legend, why the authors write: “ICP analysis data Statistics of negative graphite metal elements of lithium battery”?
I could understand if an uncertainty and confidence level were presented but was not the case.
Thank you for important comment. Because silicon and graphite in contact with some metals for a long time during early use. We and this mean to show the content level of residual metals in silicon and graphite treated by chemical process. However but we are sorry, we not performed the confidence level analysis.
(11)Pag. 11 Please review the sentence: “In this study, the SiC synthesis temperature was approximately 1550±50 °C (Figure 3. PV panels and waste graphite) are much lower than those of the traditional process.”
Thank you for your suggestion We modified it according to your suggestion.
(12)Pag. 11 Can you please indicate a reference to justify the 72% CO2 reduction, please: “…as used in this method, with corresponding CO2 releases scaled around 72% using this new method.”
Thank you for your suggestion. The CO2 emission reduction obtained in this paper is only the theoretical basis of direct emission reduction calculated from the perspective of chemical synthesis. Because there are many carbon emission nodes involved in the whole production process of traditional silicon carbide, limited by the length of the article and the focus of this paper, and we did not calculated the emission reduction of the whole production process of silicon carbide. The data calculated in this paper can be only preliminarily conclude that using the above two wastes to replace traditional raw materials, and also has certain carbon reduction value and environmental protection significance). We found relevant paper and provided citation. (Transformation of waste crystalline silicon into submicro β-SiC by multimode microwave sintering with low carbon emissions, Powder Technology, Volume 322, December 2017, Pages 290-295. https://doi.org/10.1016/j.powtec.2017.09.024)
Reviewer 2 Report
The article is interesting and well organized and it addresses importantissues related to recycling.
It can be accepted. To improve the quality, please change the drawings:
Fig.1. It should be enlarged or divided because it is illegible
Fig.4. SEM micrographs, missing scale should be corrected
Fig.5. It should be enlarged or divided because it is illegible
Author Response
The article is interesting and well organized and it addresses important issues related to recycling. It can be accepted. To improve the quality, please change the drawings:
(1)1. It should be enlarged or divided because it is illegible
Thank you for your suggestion. We improved it according to your suggestion, please see the revised Figure 1.
(2) 4. SEM micrographs, missing scale should be corrected.
Thank you for your suggestion. We updated according to your suggestion, please see the Figure 4.
(3) 5. It should be enlarged or divided because it is illegible
Thank you for your suggestion. We modified it according to your suggestion.
Reviewer 3 Report
The manuscript entitled "Microwave sintering rapid synthesis of Nano / micron β-SiC from waste lithium battery graphite and photovoltaic silicon to achieve carbon reduction", reports a study on the synthesis of silicon carbide from raw materials, such as silicon and graphite, obtained from recycling of photovoltaic solar cells and lithium batteries respectively.
The topic of the study is interesting, adequate for the purpose of the journal, but in my opinion it is mainly lacking a complete description of the methods used for the recovery of raw materials. The "Materials andmethods" paragraph should be rewritten and complete with information. For example: at what temperature were the silicon wafers burned? How was the leaching done? What is the chemical method used for graphite purification?
Furthermore, there is no analysis of the raw materials used. What is the degree of purity. A characterization of the raw materials used is necessary.
In light of the above, I believe that the work can be considered for its publication but only after revision.
Author Response
The manuscript entitled "Microwave sintering rapid synthesis of Nano / micron β-SiC from waste lithium battery graphite and photovoltaic silicon to achieve carbon reduction", reports a study on the synthesis of silicon carbide from raw materials, such as silicon and graphite, obtained from recycling of photovoltaic solar cells and lithium batteries respectively.
(1)The topic of the study is interesting, adequate for the purpose of the journal, but in my opinion it is mainly lacking a complete description of the methods used for the recovery of raw materials. The "Materials and methods" paragraph should be rewritten and complete with information. For example: at what temperature were the silicon wafers burned?
Thank you for your suggestion. We try our best to update this section according to your suggestion. Please check the revised Materials and methods parts.
(2) How was the leaching done? What is the chemical method used for graphite purification?
Thank you for your suggestion. We updated according to your suggestion, please see the Materials and methods parts.
(3) Furthermore, there is no analysis of the raw materials used. What is the degree of purity. A characterization of the raw materials used is necessary.
Thank you for comments. Since the graphite material in lithium-ion battery is high-quality graphite with high carbon content, we pay attention to the metal content after purification. In addition, the manufacturing of solar photovoltaic cells requires very high purity of silicon (about 99.9999%), and its impurities are mainly concentrated on the surface. In this study, owing to the better leaching treatment with more chemical reagents be used, some silicon on the surface of solar silicon wafer has been dissolved in alkaline solution. The silicon content in the silicon material obtained in the later stage is regarded as the material before manufacturing solar photovoltaic cells, so the silicon content is not further analyzed. We only performed the XRD analysis of Si crystal, the result has been added to the paper, please see Figure 10(right figure).
Round 2
Reviewer 3 Report
I believe that the manuscript, in its current version, can be considered for its publication